# Current Advances in the Synthesis of Valuable Dipyrromethane Scaffolds: Classic and New Methods

**DOI:** 10.3390/molecules24234348

**Published:** 2019-11-28

**Authors:** Bruno F. O. Nascimento, Susana M. M. Lopes, Marta Pineiro, Teresa M. V. D. Pinho e Melo

**Affiliations:** CQC and Department of Chemistry, University of Coimbra, Rua Larga, 3004-535 Coimbra, Portugal; nascimento@ci.uc.pt (B.F.O.N.); smlopes@uc.pt (S.M.M.L.);

**Keywords:** dipyrromethanes, dipyrromethenes, dipyrryl, BODIPY, pyrrolic macrocycles

## Abstract

This review presents the most recent developments on the synthesis of dipyrromethanes, covering classical synthetic strategies, using acid catalyzed condensation of pyrroles and aldehydes or ketones, and recent breakthroughs which allow the synthesis of these type of heterocycles with new substitution patterns.

## 1. Introduction

Dipyrromethanes are well known synthetic scaffolds for the synthesis of macrocycles and dipyrromethene metal complexes. Dipyrromethanes occupy a central place in porphyrin chemistry. The dipyrromethane structures employed in the synthesis of naturally occurring porphyrins typically bear substituents at the *β*-positions and lack any substituent at the *meso*-position. However, the dipyrromethanes with substituents at the *meso*-position have come to play a valuable role in the preparation of synthetic porphyrins [1,2], calixpyrroles [3], chlorins [4], corroles [5], and expanded porphyrins, namely saphyrins and smaragdyrins [6,7] (Figure 1).

The most representative example of dipyrromethene metal complexes are 4,4-difluoro-4-bora-3a,4a-diaza-s-indacenes also known as BODIPYs, which have been successfully used as fluorescent probes in diverse applications [8,9,10]. Other metal complexes have also attracted researchers’ attention, for example, recently, aluminium complexes have been used as catalysts for polymerization reactions [11,12], iron complexes [13] have been used as catalysts for C-H bond amination [14,15], and ruthenium complexes of dipyrromethenes [16,17] were synthetized as precursors of bis(2,2′-bipyridyl)(dipyrrinato)ruthenium(II) complexes.

Beyond its use as synthetic scaffolds, the dipyrromethane framework used as ligand for the synthesis of organometallic complexes has attracted the interest of several research groups, mostly due to the ease of synthetic accessibility and versatility of substitution of this moiety. The electronic properties of this ligand can be modified by substitution at the *beta*-carbons and at the *meso* positions, while the steric properties can be tuned by substitution at the *alpha*-carbons. Zirconium complexes have been applied in olefin hydroamination [18] and ruthenium, rhodium and iridium complexes used as hydrogen transfer catalysts under aqueous and aerobic conditions [19]. The synthesis of dipyrromethene complexes has also been achieved with Mn, Co, Zn, Ni [20,21,22] or Sn [23]. Recently, a dipyrromethane-based diphosphane–germylene was synthetized and used as precursor of tetrahedral Cu(I) and T-shaped Ag(I) and Au(I) flexible pyrrole-derived Phosphorous-Germanium-Phosphorous (PGeP) germylene pincer complexes [24].

Anion recognition is an area of growing interest due to its important role in a wide range of environmental, clinical, chemical and biological applications. Interestingly, the acidic NH protons of dipyrromethanes can be used as anion sensors with good binding activity and selectivity [25,26,27,28]. Furthermore, polymers based on dipyrromethanes were developed for the molecular recognition of two homoserine lactone derivatives involved in bacterial quorum sensing [29].

Herein, bibliographic coverage of the developments on the synthesis of dipyrromethanes since the last reviews in this area [9,30,31,32] is provided (2014–2019). The synthetic strategies have been organized in two main approaches: classical synthetic strategies based on the first report on the synthesis of *meso*-substituted dipyrromethanes, disclosed in 1974 [33], using acid catalyzed condensation of pyrrole and aldehydes; and recent breakthroughs in dipyrromethane chemistry which allow the synthesis of dipyrromethanes with new substitution patterns. 

## 2. Classic Synthetic Strategies

### 2.1. Hydrochloric Acid-Catalyzed Dipyrromethane Synthesis

Receptor molecules grounded on guanidinium- and pyrrole-containing binding sites **3** were developed by Kataev and colleagues with the objective of selective recognizing orthophosphate anions in aqueous media (Scheme 1) [27]. 

Studies demonstrated that the pyrrole-containing binding site was of pronounced influence on the selectivity and that dipyrromethane core structure **2**, prepared from the HCl-catalyzed reaction of 4-heptanone **1** and pyrrole in boiling water in 13% isolated yield, demonstrated the highest selectivity for orthophosphate over other inorganic anions. A novel and readily available dipyrromethane-based dual receptor **6** serving as colorimetric sensor for both F^-^ and Cu^2+^ ions was recently designed and prepared by Pandey and co-workers (Scheme 1) [28]. Treatment of pyrrole and acetophenone **4** in the presence of catalytic HCl in water was key in the formation of *meso*-methyl-*meso*-phenyl-dipyrromethane **5** in 75% yield.

Balci et al. established a regioselective method for the preparation of dipyrrolo-diazepine derivatives [34]. This firstly involved the classic room temperature HCl-promoted synthesis of dipyrromethanes **7** (starting from excess pyrrole and suitable aldehydes), followed by reaction of propargyl bromide **8** in the presence of sodium hydride to append an alkyne functionality to the nitrogen atom at one of the pyrrole units. A final seven-*exo*-dig cyclization, between the alkyne group and the *N*-deprotonated pyrrole moiety, followed by prototropy produced the target compounds **10** in generally good overall yields (Scheme 2).

### 2.2. Acetic/propionic Acid-Catalyzed Dipyromethane Synthesis

Interesting work carried out by the research group of Swavey allowed the preparation of two new dipyrromethane bridging ligands [17], as well as their corresponding dimetallic ruthenium(II) [17] and osmium(II) [35] coordination complexes. Reaction of phenanthrolinepyrrole **11** (php) with the selected aryl aldehyde in acetic acid (AcOH) produced dipyrromethanes **12**, comprising two php moieties linked by a *meso*-aryl group (Scheme 3). The use of benzaldehydes substituted with electron donating (vanillin) and electron withdrawing (cyano) groups did not greatly affect the efficiency of the reaction with php. However, the use of sterically hindered aldehydes, e.g., mesitylbenzaldehyde, 3,4,5-trimethoxybenzaldehyde and pentafluorobenzaldehyde, was completely unsuccessful. Coordination with Ru(II) or Os(II) bis(bipyridyl) chloride in refluxing ethanol, followed by saturation with aqueous ammonium hexafluorophosphate, created the novel dimetallic complexes **13** and **14** in good isolated yields (Scheme 3).

Access to C_2v_ symmetric *β*-substituted porphyrins, e.g., protoporphyrin III **21**, using the promptly accessible Knorr’s pyrrole **15**, which is crucial in the preparation of the required dipyrromethane building blocks [36], was envisaged by Neya and colleagues in 2016 [37]. Two Knorr’s pyrrole units were coupled into symmetric dipyrromethane **16** in propionic acid (PrOH). Its 3,3′-dibenzyl groups were removed via hydrogenolysis affording the corresponding carboxilic acid substituents, which were removed by iodination giving dipyrromethane **17**. This was further reduced to dipyrromethane **18** under a Pd/C-catalyzed hydrogen atmosphere. Reaction of **18** with acetyl chloride in the presence of aluminum chloride led to the formation of 3,3′-diacetyldipyrromethane dimethylester **19**, which was then hydrolyzed using aqueous sodium hydroxide into the corresponding dipyrromethene-5,5′-dicarboxylic acid. Its terminal carboxylic residues were subsequently eliminated through iodinative decarboxylation, rendering 5,5′-diiododipyrromethane **20** in 10.2% overall yield after eight reaction steps (Scheme 4).

The multistep synthesis of 1,4,5,8-tetraethyl-2,3,6,7-tetravinylporphyrin **26** was reported by the same research team one year later, this time using closely related Knorr’s pyrrole analogue **22** as starting material [38]. The same experimental protocol was selected in order to produce 5,5′-diiododipyrromethane **23**, which was then subjected to reduction to afford dipyrromethane **24**, followed by formylation to dipyrromethane **25** (Scheme 5). These two new dipyrromethane derivatives, **24** and **25**, were key in the subsequent preparation of the target symmetric porphyrin **26**.

### 2.3. p-Toluenesulphonic Acid-Catalyzed Dipyrromethane Synthesis

A solution of furan-2-carboxaldehyde **27** and ethyl 2-cyano-3-(1*H*-pyrrol-2-yl)-acrylate **28** in dichloromethane was refluxed for 8 h in the presence of a catalytic amount of *p*-toluenesulphonic acid (*p-*TSA) to afford 1,9-bis(2-cyano-2-ethoxycarbonylvinyl)-5-(2-furanyl)-dipyrromethane **30** in 32% yield (Scheme 6). This new dipyrromethane was extensively characterized through experimental spectroscopic measurements and theoretical quantum chemical calculations by Singh and co-workers [39]. The same authors later described the preparation of some novel dipyrromethane-hydrazone derivatives **31**, by condensing previously synthesized 2-[(4-isonicotinoyl)-hydrazonomethyl]-1*H*-pyrrole **29** with suitable aldehydes, also under classic *p-*TSA catalyzed reactional conditions, high yields being attained [40]. These were screened for antitubercular activity against *Mycobacterium tuberculosis* H37Rv strains, interesting minimum inhibitory concentration (MIC) values being found (Scheme 6).

### 2.4. Trifluoroacetic Acid-Catalyzed Dipyrromethane Synthesis

Aiming to synthesize a 5,10-diacyltripyrrane, an essential intermediary synthon for the preparation of a 5,10-diacylcalix[4]pyrrole, Mahanta and Panda were able to unexpectedly isolate acyldipyrromethane **32** with a yield up to 31% (Figure 2), following the trifluoroacetic acid (TFA)-catalyzed reaction of 2,3-butanedione and excess pyrrole [41]. Yildiz et al. have described the synthesis, characterization, crystal structure and theoretical calculations of two new *meso*-borondipyrromethene (BODIPY) incorporating a phthalonitrile moiety [42,43]. Crucial for their detailed report was the preparation of *meso*-phenoxyphthalonitrile dipyrromethanes **33** and **34** (Figure 2), which were attained in 53% and 61% yields, respectively, after typical room temperature condensation of suitable and previously obtained aldehydes with excess pyrrole in the presence of TFA. A very similar experimental setup was applied to the synthesis of unsubstituted dipyrromethane [44], *meso*-2-pyrrolyldipyrromethane **7f** (see Scheme 2) in 90% yield [45], as well as to the preparation of dipyrromethane **35**, in 43% yield [46]. Three *meso*-(trimethylsilyl)phenyl-dipyrromethane structures **36** were obtained in good yields, ranging from 60% to 66% (Figure 2), after solventless condensation catalyzed by TFA of the corresponding silylated aldehydes with pyrrole [47]. Extensive work on the synthesis of novel pentafluorosulfanyl-substituted A_4_-type porphyrins (and their respective Zn(II) and Pd(II) complexes), A_3_-, AB_2_- and A_2_B-type corroles, *trans*-A_2_B_2_-type porphyrins and BODIPYs has been reported [48]. Regarding the three latter molecular targets, *meta*-SF_5_-phenyl-substituted dipyrromethanes **37a**–**c**, prepared in high yields (62%–80%) under standard TFA-catalyzed reaction conditions using appropriate pentafluorosulfanyl-bearing aryl aldehydes and surplus pyrrole, were the strategic intermediates (Figure 2). More recently, the same authors reported the efficient preparation, under similar standard conditions, of useful *meso*-aryldipyrromethane scaffolds **37d** [49] and **37e** [50] (in 92% and 87% yields, correspondingly), which were further functionalized and/or used as building blocks for the formation of other interesting BODIPY or porphyrinoid molecular species.

Bis-dipyrromethanes **38a**–**d** and **39a**–**d**, as well as tris-dipyrromethane **38e** and **39e**, were prepared from commercially accessible starting materials by Sessler’s research team [51,52], following the strategy summarized in Scheme 7. Their anion binding properties were then evaluated in both organic media and in the solid state, compounds **39** displaying a good affinity for dihydrogenphosphate and pyrophosphate anions (as tetrabutylammonium salts) in chloroform solutions, acting as conformationally switchable receptors.

Acyclic and macrocyclic dipyrromethanes were synthesized by Love et al. by applying common Schiff-base condensation chemistry to *meso*-pentafluorophenyldipyrromethane dialdehyde **40**, diiminodipyrromethane derivative **41** being obtained in 74% yield (Scheme 8) [53]. Bridged macrocyclic dipyrromethanes **43** were prepared by condensation of **40** with either *ortho*-phenylene **42a** or 1,5-anthracene-diamine **42b**, again using TFA as catalyst. After neutralization with triethylamine, the target molecules precipitated neatly from the reaction medium (Scheme 8).

Novel bis-dipyrromethane derivatives **45** were synthesized in moderate yields by reacting previously prepared dialdehydes **44** with surplus pyrrole in the presence of TFA as catalyst (Scheme 9). These bis-dipyrromethanes **45**, as well as **38c** (see Scheme 7) and dipyrromethane **35** (see Figure 2), undergo electropolymerization on the electrode surface occurring upon multiple oxidation cycles [46]. The authors uncovered that quicker electropolymerization rates arise when the monomeric species contains more than one dipyrromethane component, and that the resulting polymers exhibit greater stability, while also showing low roughness and a very uniform and homogenous morphology.

β-Formyl-tetrapyrrolic macrocycles have been used for the synthesis of dipyrromethene derivatives through condensation with pyrroles (Scheme 10). Temelli and Kalkan recently described the synthesis of *meso*-porphyrinyldipyrromethane **47** in high yield, by reacting 2-formyl-*meso*-tetraphenylporphyrin **46** prepared beforehand and excess pyrrole under classic TFA catalysis conditions. The unforeseen construction of *β*/*meso*-directly connected diporphyrinic molecules in reactions of *β*-formylated porphyrins with pyrrole under ordinary Adler-Longo conditions was also established, early mechanistic studies showing that dipyrromethane **47**-type intermediates are fundamental in the process [54]. On the other hand, Galium(III) corrole-BODIPY hybrid **49** was obtained from the TFA-catalyzed reaction of formyl-corrole **48** with 2,4-dimethylpyrrole followed by oxidation and complexation with boron trifluoride [55].

New *meso*/*meso*-straightly linked Ni(II) porphyrin hybrids were designed and prepared by Kong and co-workers, the metalloporphyrinic units being connected with dipyrromethene, *bis*-dipyrromethene or thiacorrole units [56]. Instrumental in the synthetic strategy of the authors was *meso*-dipyrromethane-susbtituted Ni(II) porphyrin **51**, which was effortlessly obtained in good yield via TFA-catalyzed room temperature condensation of Ni(II) 10,20-di(3,5-di-*t*-butylphenyl)-5-formyl porphyrin **50** and pyrrole (Scheme 11).

The synthesis and characterization (structural, spectral and electrochemical) of Pd(II), Re(I) and Ru(II) 3-pyrrolyl-BODIPY/dipyrromethenes complexes **55a**–**c** was reported by Ravikanth’s research team in 2014 [57]. The pertinent and necessary dipyrromethane-substituted 3-pyrrolyl BODIPY intermediate **53a** was prepared in 85% yield by mixing a dichloromethane solution of formylated 3-pyrrolyl BODIPY **52a**, produced and isolated beforehand, and excess pyrrole under TFA-catalyzed and inert atmosphere conditions (Scheme 12). Related α-dipyrromethanyl 3-pyrrolyl BODIPY **53b** was synthesized in a similar fashion by the same research group a few years later (Scheme 12) [58]. This was further transformed into 3-pyrrolyl BODIPY/BODIPY dimer **54**, comprising an ethynyl functionality at the *meso*-aryl location, which was then coupled to selected monomeric BODIPYs in order to create the authors’ target near-infrared emitting BODIPY oligomers. 

### 2.5. Boron Trifluoride Diethyl Etherate-Catalyzed Dipyrromethane Synthesis

New *meso*-phenothiazinyldipyrromethanes **56**, having alkyl groups of growing bulkiness linked to the heterocyclic nitrogen of the phenothiazine core were synthesized in high yields (81%–89%) by condensing proper *N*-alkyl-phenothiazin-3-carbaldehydes with pyrrole at room temperature, in the dark, and in the presence of boron trifluoride diethyl etherate (BF_3_.Et_2_O) as catalyst (Figure 3) [59]. A BODIPY dye [59], a *trans-*A_2_B_2_-type porphyrin [59], and some Sn(IV) coordination complexes [60] bearing an *N*-methyl-phenothiazinyl motif were later prepared utilizing dipyrromethane **56a** as the building block. Thilagar and colleagues conveyed the simple preparation, under typical BF_3_.Et_2_O-catalyzed reaction conditions of four novel triarylborane-dipyrromethane derivatives **57a**–**d** (Figure 3) that encompassed dual receptor sites (Lewis acidic boron and hydrogen bond donor NH) and displayed a discriminating fluorogenic response towards the fluoride anion in dichloromethane solution [61]. Broadly acknowledged *meso*-substituted dipyrromethanes **7c** (Scheme 2) and **58** were recently obtained by Bagherzadeh and co-workers via the dropwise addition of pyrrole to a dilute aqueous solution of the aryl aldehydes, using boron trifluoride diethyl etherate as catalyst (Figure 3) [62]. This methodology was adapted from an earlier report that used aqueous HCl at 90 °C [63], tripyrromethane and other oligomers being obtained as byproducts when large and sterically hindered aryl aldehydes are employed. The reaction occurred smoothly in mild conditions, 30–70 min at 70–80 °C under an argon atmosphere, moderate to high yields (60%–85%) being obtained, even when employing bulky electron donating (mesityl) or electron withdrawing (2,6-dichlorophenyl) aldehydes, and no decomposition or scrambling being noticed. 

Sessler and co-workers synthesized pyrene-bridged bis-dipyrromethane **60** in 41% yield by reacting previously prepared pyrene dialdehyde **59** with excess pyrrole in ethanol at room temperature for three days and using BF_3_.Et_2_O as catalyst (Scheme 13) [64]. After formylation under standard reaction conditions, pyrene-linked tetraformylated bis-dipyrromethane **61** was obtained in moderate yield. Anion recognition studies revealed that the latter performs as a selective fluorescent probe in chloroform solution for dihydrogen phosphate over other tested anions.

Ravikanth’s research group reported the preparation and characterization (structural, photophysical and electrochemical) of *β-meso* covalently linked azaBODIPY/BODIPY dyad **64** [65] and Pd(II) azaBODIPY/dipyrromethene complex **65** [66,67]. The unconditionally required dipyrromethane-substituted azaBODIPY intermediary **63** was synthesized in reasonable yield by stirring a dichloromethane solution of 2-formyl azaBODIPY **62**, and surplus pyrrole, at room temperature, under boron trifluoride diethyl etherate catalysis and inert atmosphere conditions (Scheme 14). A similar synthetic strategy, condensation of 3-formyl-BODIPY with pyrrole catalyzed by BF_3_-OEt_2_ was used for the synthesis of BODIPY/BODIPY dimers [68].

### 2.6. Indium(III) Chloride-Catalyzed Dipyrromethane Synthesis

*meso*-Substituted dipyrromethanes **66a**,**b** were prepared by simple solvent-free condensation of the appropriate aryl aldehydes with excess pyrrole, under a saturated argon atmosphere and indium chloride (InCl_3_)-catalyzed conditions, (Figure 4) moderate yields being attained [69]. Lindsey’s research team designed and synthesized a series of interesting *trans*-AB-kind porphyrins and metalloporphyrins comprising one water-solubilization moiety and one bioconjugatable functionality [70]. Key for their strategy was the previous preparation of suitable dipyrromethane building blocks, including novel compound **66c**, which was obtained in 33% yield using the same catalytic conditions (Figure 4). A related setting was also applied for the synthesis of *meso*-nonyldipyrromethane **67** in 78% yield and *meso*-methoxycarbonyldipyrromethane **68** in 52% yield [71]. In addition to the latter, the synthetic process also provided regioisomer **69** and cyclic derivative **70**, a by-product resulting from intramolecular aminolysis of ester **68**, in 32% and 10% isolated yields, respectively (Figure 4). Moreover, the same authors prepared dipyrromethane derivatives **71** and **72** in moderate to reasonable yields, 47%–62% by simply mixing the adequate aldehydes or acetals with excess pyrrole in an inert atmosphere using indium chloride as catalyst. Dipyrromethanes **67–69** and **71–72** were later crucial for the preparation of several *trans*-AB-type porphyrins and metalloporphyrins (Figure 4) [71].

Aiming to combine porphyrin, BODIPY and triptycene chemistry, Senge et al. recently presented *meso*-triptycenyldipyrromethane synthon **74**, synthesized in 60% yield from the condensation of 2-formyltriptycene **73** and surplus pyrrole under standard InCl_3_-catalyzed conditions (Scheme 15) [72]. The extreme utility of dipyrromethane **74** was noticeably demonstrated by its application in the preparation of triptycene-substituted BODIPY **75a**, triptycene-substituted 3-pyrrolyl-BODIPY **75b**, *trans*-A_2_B_2_ triptycenylporphyrins **76a**,**b** and A_3_B-type triptycenylporphyrins **76c**,**d**.

Following a similar synthetic approach, the same authors also devised the synthesis of a more complex tris-dipyrromethane-substituted triptycene **78** in 37% yield [72], starting from 2,6,14-tri(4-formylphenyl)triptycene derivative **77**, which was prepared and isolated beforehand (Scheme 16). Tris-dipyrromethane **78** was later successfully utilized as a valuable intermediate in the synthesis of tris-BODIPY-substituted triptycene **79**.

### 2.7. Other Strategies

The synthetic route that Trofimov’s research group developed in order to obtain new *meso*-trifluoromethyldipyrromethanes **85** and **86** using 2-aminophenyl-1*H*-pyrroles **80** as starting material is depicted in Scheme 17 [73]. Briefly, the protection of the amino functionality of pyrroles **80** with acetic anhydride, followed by a reaction with trifluoroacetic anhydride (TFAA), rendered 2-trifluoroacetylpyrroles **82** in high yields. Sodium borohydride-promoted reduction of pyrroles **82** and ensuing reaction of pyrrole carbinols **83** with 2-phenylpyrrole **84,** in the presence of dehydration agent phosphorous pentoxide (P_2_O_5_), gave dipyrromethanes **85** in good yields. Finally, conversion of the acetamide substituents into amino groups in refluxing acidic media originated dipyrromethanes **86** (Scheme 17). Both dipyrromethane derivatives **85** and **86** were later used in the preparation of their corresponding *meso*-CF_3_-BODIPY dyes, a big influence of the coexistence of the strong electron donor NH_2_ and electron acceptor CF_3_ groups, along with the location of the amine at the aryl ring, being determined on the optical properties of chromophores **86** [73].

The preparation of novel *meso*-trifluoromethyldipyrromethanes **92** and **93**, comprising isoxazole substituents in their molecular structure, starting from ethynylpyrrole **87** was also recently described by the same authors (Scheme 18) [74]. Initial cyclization of **87** with hydroxylamine hydrochloride and subsequent condensation of the obtained isoxazoles **89** or **90** with previously prepared pyrrole carbinols **91** in the presence of dehydration agent P_2_O_5_ leads to the desired and unreported dipyrromethane derivatives **92** and **93** (Scheme 18). The latter were again further explored in the synthesis of their respective *meso*-CF_3_-BODIPY dyes, some photophysical studies and quantum chemical calculations having been carried out [74].

Aiming to synthesize the illusive and highly sought 2,3,7,8,12,13,17,18-octafluoroporphyrin with a reasonable yield, Chang and colleagues choose the approach summarized in Scheme 19 [75], ensuing an older account by Clezy and Smythe [76]. In brief, tetrafluorinated dipyrromethane **97** was obtained in three steps starting from 3,4-difluoropyrrole **94**. Reaction with thiophosgene under an inert atmosphere rendered dipyrrothioketone **95** in high yield. Subsequent hydrogen peroxide-promoted oxidation to dipyrroketone **96**, followed by sodium borohydride-mediated reduction, originated dipyrromethane derivative **97**. Having this valuable scaffold in hands, the authors were thus able to prepare *trans-*A_2_B_2_-type porphyrins **98**, as well as *β*-octafluoroporphyrins **99**.

## 3. Novel Synthetic Strategies

### 3.1. Dipyrromethane Synthesis from Aldehydes and Pyrroles

Despite the wide variety of standard available methods, there is still an open door for the development of new synthetic approaches for dipyrromethanes. For instance, there is a growing interest in the use of catalysts that can be easily removed from the reaction medium and reused, while also having an economically and ecologically friendly access. Konar and co-workers described the synthesis of a wide range of *meso*-thienyldipyrromethanes **101** using an amine functionalized MOF (Metal-Organic Framework) for the controlled release of the catalyst, iodine (Scheme 20) [77]. Dipyrromethanes **101** were obtained in high yields (50%–69%) with the ratio aldehyde/pyrrole (1:5) in the presence of 20 mol% of NH_2_-MOF(I_2_), without organic solvents and under mild conditions. The catalyst was reused for three cycles, although with a slight yield decrease. After immersion in an iodine solution, the catalytic performance was comparable with the freshly prepared NH_2_-MOF(I_2_). The authors tested other catalysts, such as unfunctionalized MOFs (H-MOF(I_2_)), conventional TFA and molecular iodine; however, dipyrromethanes **101** were obtained with lower yields.

Copper nanoparticles (CuNPs) [78] and celite-supported glycine (glycine@celite) [79] have been explored as catalysts in the synthesis of dipyrromethanes because they are easily removed from the reaction medium and can be reutilized (Scheme 21). The reaction of aromatic aldehydes with pyrrole (10 equiv) under solvent-free conditions using CuNPs as catalyst gave dipyrromethanes **7c**,**e**, **58d** and **103** in high yields (65%–77%). In this catalytic system, the pyrrole nucleophilicity was increased by the adsorption over CuNPs and reacted with the electrophilic carbon of the aldehyde to give an intermediate, which reacted with another molecule of pyrrole adsorbed on CuNPs, giving dipyrromethanes **7c**,**e**, **58d, 103**. The catalyst was recovered and reused in four cycles with a similar catalytic activity without any deterioration. Dipyrromethane **104d** was also synthesized using the CuNPs as catalyst in 75% yield (Scheme 21) [78]. Glycine@celite was a milder acid catalyst system whose efficacy was proven in the synthesis of dipyrromethanes **104**. The reaction of aromatic aldehydes with 2,4-dimethylpyrrole (3 equiv) in the presence of 10 mol% of the catalyst in dichloromethane for 30 min gave the corresponding dipyrromethanes **104** in very good yields (Scheme 21). The authors described the reuse of the catalyst for 5 cycles without any deterioration and with similar catalytic activity [79].

Dipyrromethane **104e** underwent oxidation and chelation by copper producing a red bis(dipyrrinato)copper(II) complex **105** (Scheme 22). This characteristic makes it an efficient naked eye colorimetric chemosensor for copper ions, even in the presence of several other metal ions [79,80].

Dipyrromethanes **7c**, **103b** and **106** were synthesized in good yields by the iodine-catalyzed double Friedel-Crafts reaction, using toluene or water as solvent (Scheme 23) [81]. The reaction of pyrrole derivatives with aldehydes, (2:1) ratio, in presence of 10 mol% of molecular iodine in water gave dipyrromethanes in better yields (60%–87%) than the reaction carried out in toluene (42%–62%).

Sirion and co-workers described the synthesis of dipyrromethanes through the reaction of aldehydes with pyrrole catalyzed by SO_3_H-functionalized ionic liquids (SO_3_H-ILs) in aqueous media [82]. The authors tested a few SO_3_H-ILs containing imidazolium or pyridinium cations and different anions, and found that [bsmim][HSO_4_] (i.e., 1-butylsulfonic-3-methylimidazolium hydrogen sulfate) was the ideal catalyst for the synthesis of dipyrromethanes (Scheme 24). Dipyrromethanes **107** were obtained in moderate to good yields from the reaction of pyrrole with aliphatic or aromatic aldehydes, in the presence of 10 mol% of catalyst in water under mild conditions. The described method comprised a large variety of aromatic aldehydes with both electron withdrawing and electron donating substituents, heteroaromatic aldehydes as well as alkyl aldehydes. Moreover, the catalyst was easily removed from the reaction media trough a simple extraction and recycling [82]. Later, the synthesis of *meso*-aryldipyrromethanes using 1-propylsulfonic-3-methylimidazolium trifluoromethylacetate as a catalyst was described, however, organic solvents were used [83].

The research group of Majee explored the use of imidazolium zwitterionic molten salt as organocatalyst in the synthesis of *meso*-substituted dipyrromethanes under solvent-free conditions (Scheme 25) [84]. This catalytic system acts as an electrophilic activator of the aldehyde by the hydrogen bond with imidazolium C-2 hydrogen, emphasizing the importance of this cationic moiety. The reaction of pyrrole or *N*-methylpyrrole with aromatic or aliphatic aldehydes (ratio 2:1), in the presence of 10 mol% of the catalyst at room temperature and without solvent, gave access to a wide range of dipyrromethanes in very high yields. Aromatic aldehydes containing electron withdrawing or donating groups react with pyrrole or *N*-methylpyrrole to give dipyrromethanes **108a** in yields from 72% to 87%. Dipyrromethanes with a naphthyl **108b**, pyrrole or indole **108c**, and propyl **108d** substituents were also synthesized in high yields, using the same imidazolium zwitterionic molten salt as catalyst. The methodology developed was seen by the authors as being a green synthetic protocol, because it was metal- and solvent-free, and environmentally friendly with a good atom economy [84].

*meso*-Acetyldipyrromethane **110** was synthesized in 70% yield by the reaction of methylglyoxal **109** with pyrrole, in a 1:2.5 ratio, using boric acid as catalyst in aqueous media (Scheme 26) [85]. In addition to dipyrromethane **110**, other dipyrromethanes were synthesized using aromatic aldehydes and similar reaction conditions. Boric acid is weakly acidic and reacts with water decreasing the pH of the aqueous layer; given that the reaction of pyrrole with aldehydes occurs in the interface with the organic layer, the formation of side products is thus prevented.

### 3.2. Dipyrromethane Synthesis via Alternative Methods

Pinho e Melo and co-workers developed an on-water one-pot synthetic approach to *meso*-substituted dipyrromethanes via hetero-Diels-Alder reaction (or conjugated addition) of nitrosoalkenes and azoalkenes with pyrrole (Scheme 27) [86,87]. Dehydrohalogenation of α,α-dihalooximes or α,α-dihalohydrazones, in the presence of base, produces transient nitrosoalkenes or azoalkenes **I**. These reactive species react with pyrrole to give pyrroles **II** functionalized at C-2 with a side chain, which undergo dehydrohalogenation to form the second transient nitrosoalkenes or azoalkenes **III**. The reaction with another molecule of pyrrole gives the dipyrromethanes **112** or **113**, in moderate to high yields (21%–82%). The formation of dipyrromethanes **112** and **113** are accelerated and more efficient using water as solvents allowing easier purification procedures than the reaction performed in dichloromethane or in the absence of solvent. Dipyrromethanes synthesized by this approach have the unique feature of being *meso*-substituted with oxime and hydrazone moieties [86]. The same research group described the functionalization of dipyrromethanes at positions 1 and/or 9 through hetero-Diels-Alder reaction or conjugated addition of nitrosoalkenes and azoalkenes [88,89,90]. 

The one-pot synthesis of *ortho*-hydroxymethyl 8-C-aryl BODIPY derivatives **117** was achieved through the key intermediate dipyrromethanes **116** (Scheme 28) [91]. Ethyl phthalidinium salts **115**, obtained by *O*-ethylation of phathalides **114** using Meerweins reagent, reacted with pyrrole to form intermediate ketal **I**. Elimination of the ethoxy group from intermediate **I** gave the oxonium ion **II**, which reacted with a second pyrrole unit to produce dipyrromethane **116**. Reaction of dipyrromethanes **116** with BF_3_.OEt_2_ gave the BODIPY derivatives **117** in moderate yields (26%–45%). The masked 5-alkoxy-5-phenyldipyrromethane **116** with R^1^ = H, was isolated and treated with BF_3_.OEt_2_ in order to confirm that this is a key intermediate in the synthesis of the corresponding borondipyrromethenes **117**.

Borbas and Xiong developed a strategy to synthesize unsymmetrical dipyrromethanes **120** through the Mannich reaction between pyrroles and Eschenmoser’s salt (Scheme 29) [92]. Initially, pyrroles **118** reacted with Eschenmoser’s salt to give the Mannich product **119**, which undergo substitution of the *N*,*N*-dimethylamino group under microwave irradiation using pyrrole as reactant and solvent. This method encompasses acid-sensitive and formyl groups and does not require the use of acid to activate the pyrrole unit.

*meso*- and *α*-Unsubstituted dipyrromethanes **124** were formed by the decarboxylation of 1,9-diethoxycarbonyldiyrromethanes **123** with KOH in ethylene glycol (Scheme 30) [93]. Bromination of the α-methyl group of pyrrole **121**, followed by nucleophilic substitution generated α-acethoxymethyl pyrroles **122**, which underwent self-condensation in the presence of HCl to give *meso*-unsubstituted dipyrromethanes **123** in good yields. Dipyrromethanes **124** are key intermediates in the synthesis of porphyrins **125**, that are *meso*-unsubstituted and *β*-substituted.

Thompson and co-workers developed a methodology to generate dipyrromethanes through the microwave-assisted reduction of *F*-BODIPYs and dipyrromethenes (Scheme 31) [94]. *meso*-Aryl BODIPYs **126** or dipyrromethenes **127** are reduced to the corresponding dipyrromethanes **128** in ethylene glycol and an excess of sodium methoxide under microwave irradiation at 215 °C for 10 minutes. This methodology is useful when BODIPYs or dipyrromethenes are formed in one-pot procedures and it is necessary to regenerate the dipyrromethane.

Neo-confused porphyrins **133** have been synthesized from the reaction of neo-confused dipyrromethanes **131** with dipyrromethane **132** (Scheme 32) [95]. The treatment of pyrrole-3-carboxaldehyde **129** with NaH in DMF at room temperature, followed by addition of methyl 4-formylpyrrole-2-carboxylate (**130**) gave the corresponding neo-confused dipyrromethanes **131** in good yields (45%–75%).

## 4. Conclusions

Given the continual relevance of the dipyrromethane scaffold, either by its own merits and applications or because of its extreme usefulness as a synthetic intermediate for other high value molecules, e.g., calix[4]pyrroles, (hydro)porphyrins, expanded porphyrins, corroles, BODIPY dyes, and metal coordination compounds, classical synthetic methods employing effective tried-and-tested catalysts still find their place on laboratory benches across the world. Nonetheless, as can be realized from this literature review covering the past six years, it is highly expected that organic and medicinal chemists, as well as material scientists, will keep pursuing innovative technologies and/or novel synthetic approaches with the aim of obtaining original, interesting, and much needed dipyrromethane structures.

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
