# Peer review of "Current Advances in the Synthesis of Valuable Dipyrromethane Scaffolds: Classic and New Methods"

_molecules, 2019, doi:10.3390/molecules24234348_

Round 1
Reviewer 1 Report
The article is adequated, and since 2016 there has not been published a review on this field.
I just found this items in 2019
Rational Design of Ratiometric Near-Infrared Aza-BODIPY-Based Fluorescent Probe for in Vivo Imaging of Endogenous Hydrogen PeroxideQuick View Other Sources By Mao, Wenle; Zhu, Mingming; Yan, Chenxu; Ma, Yiyu; Guo, Zhiqian; Zhu, Weihong From ACS Applied Bio Materials (2019), Ahead of Print. | Language: English, Database: CAPLUS
Precise in vivo tracking of hydrogen peroxide is still challenging due to its dynamic complexity and intrinsic background interference. Herein, we describe a rational design strategy to construct asym. aza-boron-dipyrromethane deriv. (BODIPY)-based ratiometric probes for in vivo tracking H2O2, which are composed of a near-IR aza-BODIPY core, active targeting group, and H2O2-specific recognition unit. We take advantage of two terminal functionalized conjunctions in the bis-condensed aza-BODIPY by rationally introducing carbonyl group as an electron-deficiency linker for regulating intramol. charge transfer-induced wavelength shift and by attaching hydrophilic polyethylene glycol-biotin segment as the active targeting moiety. The probe BP5-NB-OB features several striking characteristics: (i) ratiometric near IR response in both absorption and emission spectra; (ii) active targeting ability (biotin receptor-mediated endocytosis) with excellent biocompatibility; and (iii) in vivo tracking of endogenous H2O2. It was demonstrated that the probe BP5-NB-OB was successfully utilized for tracking endogenous H2O2 in living cells and tumor-bearing mice, providing opportunities to insight into H2O2 related diseases for clin. application.
|
~0 |
2.
Synthesis and X-ray single crystal study of 5-(4,4,5,5 - tetramethyl - 1,3,2 - dioxoborolane) - 10,20 - diphenylporphyrinQuick View Other Sources By Radzuan, Nuur Haziqah Mohd.; Abdul Malek, Nawwar Hanun; Ngatiman, Mohammad Fadzley; Xin, Tan Ke; Bakar, Mohd. Bakri; Hassan, Nurul Izzaty; Abu Bakar, Muntaz From Sains Malaysiana (2018), 47(9), 2083-2090. | Language: English, Database: CAPLUS
Borylated porphyrin is one of building blocks in coupling reactions to obtain the multiporphyrin contg. two, three or more subunits of porphyrins. In this study, one of borylated porphyrin derivs., 5-(4,4,5,5 - tetra-Me - 1,3,2 - dioxoborolane) -10,20 - diphenylporphyrin (B-dpp) was synthesized through four steps of reactions. The building block of porphyrin, dipyrromethane was synthesized through a condensation reaction in the presence of trifluoroacetic acid as catalyst. Subsequently, A2B2 type of porphyrin was obtained by Lindsey condensation reaction followed by bromination reaction to produce porphyrin halide. Suzuki cross coupling reaction between porphyrin halide and pinacolborane with Pd (II) catalyst afforded 40% of borylayed porphyrin. The product was successfully characterized by using NMR spectroscopy (nmr) and uv-Visible spectroscopy (uv-Vis). This compd. crystd. from a mixt. of dichloromethane/methanol to give violet needle-like crystal. Crystallog. studies showed this compd. crystd. in monoclinic system with space group of P21/c.
|
~0 |
3.
Chemical varactor-based sensors with non-covalent, electrostatic surface modification of grapheneQuick View |
By Zhen, Xue; Buhlmann, Philippe Pierre Joseph; Koester, Steven J.; Nelson, Justin Theodore From PCT Int. Appl. (2019), WO 2019209918 A1 20191031. | Language: English, Database: CAPLUS
A medical device including a graphene varactor, including a graphene layer and a self-assembled monolayer disposed on an outer surface of the graphene layer through electrostatic interactions between a partial pos. charge on hydrogen atoms of one or more hydrocarbons of the self-assembled monolayer and a pi-electron system of graphene. The self-assembled monolayer includes one or more substituted porphyrins or substituted metalloporphyrins, or derivs. thereof. A method of fabricating the graphene varactor as well as a method of det
that have not been covered, but it is not an inconvenienceAuthor Response
The suggested reference 2 was added to the manuscript as ref 44, line 162
Reference 1 and reference 3 were not included in the manuscript. Ref 1 is an interesting work about aza-BODIPY but this type of derivatives this type of derivatives is not within the scope of this manuscript. In reference 3 (patent), there no examples describing the synthesis of dipyrromethanes.
Reviewer 2 Report
The authors have written a very good review on the synthesis of dipyrromethanes. It covers the most recent developments of their synthesis since the last reviews in this area is provided. Throughout the review, the authors present many useful examples of the preparation of dipyrromethane derivatives, which will be interest to a large number of readers in the field of porphyrin chemistry. Therefore, I recommend publication of this paper as is.
Author Response
We thank the reviewer for his/her positive comment regarding our manuscript.
Reviewer 3 Report
This review article by Prof. Melo and co-workers intends to compile various methodologies reported for the synthesis of dipyrromethane. Undeniably, dipyrromethane plays a central role in the porphyrinoids chemistry and also the cornerstone of accelerated developments in the area of BODIPYs. Considering its diversity in reactivity and structures, the current review had restricted itself a little. I feel, many important and relevant developments could have been included to expand the horizon and usefulness of the article. For example, synthetic achievements in N-confused dipyrromethane could be discussed in connection to their regular dipyrromethane. In addition, there are numerous unique examples where dipyrromethanes have been constructed on various chromophoric scaffolds (e.g., Org. Lett. 2015, 17, 21, 5360-5363; Chem. Eur. J. 2017, 23, 4837−4848; Dalton Trans., 2015, 44, 20817) leading to hybrid molecules with unique properties. Thus, incorporating such development would definitely enhance the visibility of this review article.
There are many typos errors which need to be corrected.
(a) ref. 1, 44, 53 and in many places, please correct as b or meso (all in italics).
(b) ref. 5 and 31, correct as Orłowski and Gryko.
(c) ref. 9, 32, 48, 55, 56 and in many places, correct as BODIPY/BODIPYs.
(d) ref. 14 and 15, correct as C-H
(e) ref. 20, correct as Ni(II) or Ni(ii). Similarly, in ref. 75 Cu(II)
(f) ref. 28, correct as F- and Cu2+
(g) ref. 47, correct as SF5, 8-CF3-BODIPY (ref. 69) and SO3H (ref. 78)
(h) ref. 50, correct as 1H-pyrrole-2-carboxyldehyde
(i) ref. 64, correct as Pd(II)
(j) ref. 67, correct as trans-AB-porphyrin
(k) ref. 90, correct as F-BODIPYs
Author Response
The following references suggested by the reviewer were included in the manuscript:
Chem. Eur. J. 2017, 23, 4837−4848 included as reference 68, lines 297-298
Dalton Trans., 2015, 44, 20817 included as reference 55, Scheme 10 lines 221-224
Type errors were corrected.
Reviewer 4 Report
This review presents the most recent developments on the synthesis of dipyrromethanes by classical synthetic strategies, starting from pyrroles and aldehydes or ketones using acid catalyzed condensation, and recent breakthroughs which allow the synthesis of these type of heterocycles with
new substitution patterns.The review is quite complete, I suggest to the authors to insert a recent paper (August 2019) published on J. Org. Chem. 2019, 84, 17, 10775-10784 by Angira Koch and Mangalampalli Ravikanth. In the references check the number 9, 29 and 54.
Author Response
The reference suggested by the reviewer (J. Org. Chem. 2019) was added, see line 293
References 9, 29 and 54 were checked. Type errors were corrected.